# Patient feedback surveys among pregnant women in Eswatini to improve antenatal care retention

Chloe A. Teasdale[1,2,3]*, Amanda Geller[4], Siphesihle Shongwe[2], Arnold Mafukidze[2], Michelle Choy[2], Bhekinkhosi Magaula[2], Katharine Yuengling[2], Katherine King[5], Eduarda Pimentel De Gusmao[2], Caroline Ryan[6], Trong Ao[6], Tegan Callahan[4], Surbhi Modi[4], Elaine J. Abrams[2,3,7]

1 Department of Epidemiology & Biostatistics, CUNY Graduate School of Public Health & Health Policy, New York, NY, United States of America, 2 Mailman School of Public Health, ICAP-Columbia University, New York, NY, United States of America, 3 Department of Epidemiology, Mailman School of Public Health, Columbia University, New York, NY, United States of America, 4 US Centers for Disease Control and Prevention, Atlanta, GA, United States of America, 5 NYC Department of Health and Mental Hygiene, NYC Health Training—Clinical Operations and Technical Assistance Program, New York, NY, United States of America, 6 US Centers for Disease Control and Prevention, Mbabane, Eswatini, 7 Department of Pediatrics, Vagelos College of Physicians & Surgeons, Columbia University, New York, NY, United States of America

* chloe.teasdale@sph.cuny.edu

**Data Availability Statement:** The data from the anonymous patient feedback survey is provided as Supporting Information. The patient level data used for the retention analysis is available by request.

## Abstract

### Background

Uptake and retention in antenatal care (ANC) is critical for preventing adverse pregnancy outcomes for both mothers and infants.

### Methods

We implemented a rapid quality improvement project to improve ANC retention at seven health facilities in Eswatini (October-December 2017). All pregnant women attending ANC visits were eligible to participate in anonymous tablet-based audio assisted computer self-interview (ACASI) surveys. The 24-question survey asked about women's interactions with health facility staff (HFS) (nurses, mentor mothers, receptionists and lab workers) with a three-level symbolic response options (agree/happy, neutral, disagree/sad). Women were asked to self-report HIV status. Survey results were shared with HFS at monthly quality improvement sessions. Chi-square tests were used to assess differences in responses between months one and three, and between HIV-positive and negative women. Routine medical record data were used to compare retention among pregnant women newly enrolled in ANC two periods, January-February 2017 ('pre-period') and January-February 2018 ('post-period') at two of the participating health facilities. Proportions of women retained at 3 and 6 months were compared using Cochran-Mantel-Haenszel and Wilcoxon tests.

### Results

A total of 1,483 surveys were completed by pregnant women attending ANC, of whom 508 (34.3%) self-reported to be HIV-positive. The only significant change in responses from

Per the approved study protocol, all routinely-collected electronic medical record data belong solely to the Eswatini Ministry of Health. Requests for access to these data can be sent to ICAP at Columbia University, ct116@columbia.edu.

**Funding:** Yes, the authors listed from ICAP at Columbia University received funding from the US Centers for Disease Control and Prevention (U2GGH00994) to complete the work described in the paper. All funding was received by Columbia University (no individual funding).

**Competing interests:** The authors have declared that no competing interests exist.

**Abbreviations:** ANC, Antenatal care; ART, Antiretroviral therapy; DHS, Demographic and Health Surveys; ECMIS, Eswatini Client Management Information System; HFS, Health facility staff; HIV, Human immunodeficiency virus; IQR, Interquartile range; PMTCT, Prevention of mother-to-child transmission (of HIV); PPRS, Patient Provider Relationship Scale; RLS, Resource limited setting; WHO, World Health Organization; WLHIV, Women living with HIV.

month one to three was whether nurses listened with agreement increasing from 88.3% to 94.8% (p<0.01). Overall, WLHIV had significantly higher proportions of reported satisfaction with HFS interactions compared to HIV-negative women. A total of 680 pregnant women were included in the retention analysis; 454 (66.8%) HIV-negative and 226 (33.2%) WLHIV. In the pre- and post-periods, 59.4% and 64.6%, respectively, attended at least four ANC visits (p = 0.16). The proportion of women retained at six months increased from 60.9% in the pre-period to 72.7% in the post-period (p = 0.03). For HIV-negative women, pre- and post-period six-month retention significantly increased from 56.6% to 71.6% (p = 0.02); however, the increase in WLHIV retained at six months from 70.7% (pre-period) to 75.0% (post-period) was not statistically significant (p = 0.64).

## Conclusion

The type of rapid quality improvement intervention we implemented may be useful in improving patient-provider relationships although whether it can improve retention remains unclear.

## Background

Uptake and retention in antenatal care (ANC) is critical for preventing adverse pregnancy outcomes for both mothers and infants. The World Health Organization (WHO) recommends that all pregnant women attend a minimum of four ANC visits to prevent complications which contribute to high maternal and infant morbidity and mortality in resource limited settings (RLS) [1,2]. For pregnant women living with HIV (WLHIV) it is critical that they remain in care throughout their pregnancy and after so that they receive the full package of prevention of mother-to-child transmission of HIV (PMTCT) services. In the context of Option B+, which calls for lifelong antiretroviral therapy (ART) for all pregnant and breastfeeding WLHIV, women coming to ANC should be tested for HIV and those testing positive must be rapidly initiated on ART [3]. Unfortunately, many women in RLS do not complete the minimum package of ANC services, and retention of WLHIV in PMTCT services has been a significant challenge. Data from recent Demographic and Health Surveys (DHS) from 35 sub-Saharan African countries showed that only 60.2% of women completed four ANC visits (98.1% had at least one visit) [4]. Among WLHIV, who must be retained during pregnancy and through breastfeeding, which can last 24 months or longer, only an estimated 76.4% of those entering PMTCT services are retained at six months [5].

Multiple factors help explain why many pregnant women do not complete ANC and PMTCT care, including lack of access, high costs, and poor quality of care [6,7]. Quality factors that may negatively impact pregnant women's uptake and retention in care include low skill levels of providers, lack of privacy, prolonged duration of visits, and insufficient health education [6,8,9]. Relationships between patients and providers and attitudes of providers in ANC and PMTCT settings have also been examined as factors that influence retention [10–12]. There have been reports of poor treatment of pregnant women by healthcare providers globally ranging from disrespectful attitudes to physical abuse [13–15]. The WHO's 2016 guidelines call for ANC services that are "person-centered" and provide "respectful care that takes into account woman's views" [1]. In a discrete choice experiment conducted with pregnant and post-partum WLHIV in Ethiopia and Mozambique, "respectful and pleasant providers" was ranked high on a list of desired service delivery attributes for PMTCT programs [16].

                                                                                                        

Patient-provider relationships have been shown to impact acceptance of PMTCT services in South Africa [11]. Despite these finds, few studies have examined whether efforts to improve patient-provider relationships can increase retention in ANC and PMTCT services.

We report findings from the evaluation of an intervention aimed at increasing retention of pregnant women attending ANC in the Kingdom of Eswatini (formerly Swaziland) through improving relationships with health facility staff (HFS). Eswatini has relatively high ANC coverage for a RLS, with 76% of pregnant women attending at least four ANC visits and 88% of women delivering with a skilled birth attendant [17]. While Eswatini faces a high HIV burden, with almost 35% HIV prevalence among women of reproductive age, UNAIDS estimates that 79% of pregnant WLHIV received antiretroviral medications for PMTCT in 2018 [18]. The study intervention was based on the principles of quality improvement initiatives (QI), in which health facilities use data to inform and improve services [19], and was designed to be implemented rapidly over three months. Previous studies in sub-Saharan Africa have examined QI initiatives to improve outcomes among women in ANC and PMTCT services primarily through HCW training and facility system strengthening, and have had mixed findings [20–22]. The goal of our intervention was to improve patient-provider relationships as a means to improve service uptake and retention among all pregnant women in ANC, including those living with HIV. The project included the collection of anonymous patient satisfaction surveys from pregnant women regarding their interactions with health facility staff (HFS) in the ANC. Data from the surveys were then presented to HFS at monthly feedback sessions which included discussions and exercises to help identify strategies to improve services. We evaluated changes in satisfaction over time and by HIV status and also measured retention of pregnant women using routinely collected facility data in the periods before and after the patient feedback survey intervention.

## Methods

This study implemented a rapid QI intervention at seven high volume health facilities in the Manzini region of Eswatini. The project design was informed by the QI initiatives however it was adapted to be short in duration (4–5 months) and was targeted specifically at improving patient-provider relationships through the use of patient feedback surveys and review of these data with HFS (described further below). The study included the collection of patient feedback surveys from pregnant women attending ANC services and HCW feedback sessions to share survey results and identify approaches to improving patient-provider relationships.

### Surveys

Patient feedback surveys were collected at the seven participating health facilities starting in late August or early September 2017 (depending on the site) and data collection ended at all sites on December 15, 2017. All pregnant women attending ANC at the participating health facilities were eligible and offered the opportunity to participate in an anonymous electronic tablet-based survey. Women who provided verbal consent completed the survey in the facility following a care visit. Pregnant women less than 18 years of age were asked to participate and able to provide consent for themselves as pregnant women in Eswatini are considered emancipated minors. The survey was administered on electronic tablets using audio-assisted computer self-interview (ACASI) with a three-level symbolic response option (Fig 1). Women read and/or listened (through headphones) to the survey questions and then provided a response (agreement, disagreement, or neutral) or could decline to answer any question. The 24-question survey asked about women's interactions with HFS [nurses, mothers2mothers (mentor mothers) peer supporters, receptionists and lab workers] including whether women felt

                                                            

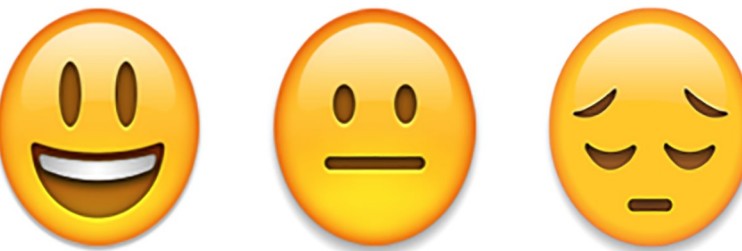

**Fig 1.**

respected and whether nurses spent enough time with them and answered their questions. All survey questions were phrased so that agreement represented favorable feelings (i.e. satisfaction) and disagreement represented unfavorable feelings (i.e. lack of satisfaction) about interactions with providers. At the end of the survey women were also asked to self-report age, HIV status, whether their home had electricity, and whether this was the first visit ANC (during the current pregnancy), with the option to decline. Women were invited to complete surveys at all visits during the three-month period; information about previous survey participation was not collected.

The patient feedback survey was designed for both HIV-positive and HIV-negative pregnant women based on the Patient Provider Relationship Scale (PPRS), a validated instrument developed in South Africa for the ANC and PMTCT context that specifically focuses on soliciting patient feedback regarding providers listening to, caring about, and respecting them [11]. The PPRS was translated into siSwati and additional questions were added (based on existing questions) to ask about interactions with non-clinical providers (mentor mothers, receptionists, and lab technicians) among the subset of women reporting interactions with these staff at the visit. Questions were not asked about specific providers so responses could not be linked to individual HFS. The tool was not formally validated following the translation and addition of new questions; however, it was field tested with local staff for comprehension.

## Feedback sessions

Quality improvement sessions were held at the end of each of the three months of survey collection with HFS to review patient survey data from their facility. Sessions were led by one nurse from each facility who received training from the study staff to facilitate discussions aimed at identifying strategies to improve patient-provider relationships. The feedback sessions utilized adapted standard QI tools [23] including run charts with data from the previous and current month patient surveys, root cause analysis and the "five whys" framework for understanding the nature of the issues identified in the surveys. Sessions also involved brainstorming for ways to improve relationships women attending ANC services. No additional resources were provided by the study to implement the improvement strategies (such as hiring additional staff or improving facilities). Formal data were not collected from the HFS meetings on the strategies identified and information is not available on what changes were planned and/or implemented.

We present the patient feedback survey response data for each question with the total number reporting and proportions of women who agreed (indicating satisfaction), disagreed (lack of satisfaction), or were neutral among those who responded. In order to estimate whether

patient satisfaction improved over time, we compared the proportions of responses (agree, neutral, disagree) between the first and third month, and we also compared the responses over all three months between women self-identifying as HIV-negative and HIV-positive. Although survey data collection was ongoing throughout the period from August/September through December 2017, some surveys were excluded from our analysis in order to examine changes in patient feedback following the HFS feedback sessions. Scheduling of HFS feedback sessions was delayed at all facilities after the first month of survey data collection and at three facilities following the second month of survey collection. We excluded surveys at each facility that were collected after the end of the first month of survey collection (>30 days) but before the HCW feedback in order to ensure that months 1 surveys included the same duration across all facilities and so that we could clearly identify data from before and after each HFS feedback session throughout the entire data collection period. Comparisons of proportions by month and HIV-status were conducted using Chi-square tests and Fisher exact tests.

In order to assess whether retention improved following the patient feedback survey intervention, data from the Eswatini Client Management Information System (ECMIS) were utilized from the two health facilities included in the project with electronic medical records. Per standard procedures at each site, data in ECMIS were entered by HFS at the time of each visit including demographic and clinical variables. All ECMIS data were de-identified prior to being given to study staff. We compared outcomes between pre- and post-period cohorts of women from the two health facilities: the pre-period cohort included pregnant women who had a first ANC visit in January and February 2017 (before the patient feedback intervention) and the post-period cohort included women newly enrolled in ANC in January and February 2018 (the period following the intervention). The two health facilities with available electronic medical records included a large government hospital and a clinic run by a non-governmental organization both located in the capital city of Mbabane. Characteristics of women at the first ANC visit including age, gestational age, trimester and self-reported HIV status are reported. We estimated the median number of ANC visits among all pregnant women and the proportion who attended at least four ANC visits. Delivery date was not recorded in the ECMIS, so ANC visits (i.e. those during pregnancy) were identified by estimating the expected date of delivery based on the gestational age at first ANC visit and assuming a 40-week pregnancy. We also calculated the proportion of all pregnant women retained in ANC care at three and six months after their first ANC visit among those with a first ANC visit <33 weeks gestation and <19 weeks, respectively (a one-month window before and after was applied for each retention endpoint). Cochran-Mantel-Haenszel (adjusted for facility) and Wilcoxon tests were used to compare differences in proportions and medians, respectively, between groups. The study protocol was reviewed and approved by the Columbia University Medical Center Institutional Review Board (IRB), and the Eswatini National Health Research Review Board. The study was also reviewed in accordance with and the US Centers for Disease Control and Prevention (CDC) human subjects protections procedures and was determined to be research, but CDC investigators did not interact with human subjects or have access to identifiable data or specimens for research purposes.

## Results

From August/September to December 2017, 1,483 surveys were completed by pregnant women attending ANC, of whom 508 (34.3%) self-reported to be HIV-positive, 710 (47.9%) reported HIV-negative status and 265 (17.9%) declined to report HIV status. The median age of participants was 25 years (interquartile range [IQR] 21–30); 35.0% completed the survey at their first ANC visit for the current pregnancy, and 82.6% reported having electricity in their

home. The questions with the highest proportion of "agree" responses overall were whether mothers2mothers peer supporters had treated the woman with respect (97.4%), whether nurses showed them care (96.3%), and whether nurses answered all questions from the participant during the clinical visit (96.2%) (Table 1). The question with the lowest agreement and highest disagreement overall was whether women were satisfied with wait times (56.0% agreed/were satisfied while 26.0% disagreed/unsatisfied). The only significant change in responses from month 1 to 3 was regarding whether nurses listened to women, for which agreement increased from 88.3% to 94.8% (p<0.01).

Overall, WLHIV had significantly higher proportions of agreement or reported greater satisfaction with HFS interactions compared to HIV-negative women; WLHIV were more likely to report that nurses talked to them about and helped them solve their problems (Table 2). Compared to HIV-negative women, WLHIV were also more likely to say that nurses supported their decisions and to report satisfaction with wait times; while 83.8% of HIV-negative women agreed that nurses supported their decisions (and 13.5% disagreed). Among WLHIV, 90.3% felt nurses supported and only 6.2% did not feel nurses supported their decisions (p<0.001) (Table 2). Overall, a high proportion of all women reported that nurses treated them with respect including 98% of WLHIV. A higher proportion of WLHIV (43.9%) compared to HIV-negative women (24.5%) reported feeling that HFS treated HIV-positive women worse than HIV-negative women (p<0.0001).

A total of 680 pregnant women from two health facilities were included in the retention analysis; of these, 454 (66.8%) were HIV-negative and 226 (33.2%) were WLHIV (Table 3). The median age of women attending a first ANC visit was 27 years in both the pre- and post-period cohorts. In the pre-period, median gestational age at first ANC was 20 weeks (IQR 15–25) and in the post-period was 19 weeks (IQR 14–24). Overall, the proportion of pregnant women attending four ANC visits increased from 59.4% in the pre-period to 64.6% in the post-period; however, the difference was not statistically significant (Fig 2; p = 0.16). The proportion of all women who had a first ANC visit at <19 weeks (N = 305) who were retained at six months increased from 60.9% in the pre-period to 72.7% in the post-period (p = 0.03). For HIV-negative women, pre- and post-period six month retention significantly increased from 56.6% to 71.6% (p = 0.02); however, the increase in the proportion of WLHIV retained at six months from 70.7% in the pre-period to 75.0% in the post-period was not statistically significant (p = 0.64).

## Discussion

The objective of this study was to evaluate an intervention to increase retention of pregnant women in Eswatini through a rapid quality improvement initiative aimed at improving patient-provider relationships in ANC and PMTCT settings. The anonymous electronic tablet-based surveys collected from all ANC attenders revealed high levels of satisfaction with their interactions with nurses, peer mentors, receptionists, and laboratory personnel, particularly among WLHIV. At baseline, the vast majority of women agreed that they had received respective treatment from mothers2mothers peer mentors (97.4% agreed), nurses (95.4% agreed), and laboratory staff (95.4%), and that nurses cared about them (96.3%) and answered their questions (96.2%). While there was little change in patient satisfaction during the three-month intervention period, there was a significant improvement in the proportion of all women reporting that nurses listened to them from the first to the third month. To measure the endpoint of retention in ANC and PMTCT services, routinely-collected patient visit data revealed that in the period prior to the intervention, only 59.4% of all pregnant women newly entering ANC in the pre-intervention period attended the recommended four antenatal visits

**Table 1. Patient feedback survey responses from pregnant women attending antenatal care at seven health facilities in Eswatini, August/September-December 2017 (N = 1,483).**

| Survey question | All surveys (N = 1483) | | | | Month 1 (N = 568) | | | Month 3 (N = 392) | | | |
|---|---|---|---|---|---|---|---|---|---|---|---|
| | Number responded | Percent responded | Percent Agree | Percent Disagree | Percent Neutral | Percent Agree | Percent Disagree | Percent Neutral | Percent Agree | Percent Disagree | Percent Neutral | p-value |
| Did your nurses listen to you? | 1443 | 97.3 | 91.8 | 5.2 | 3.0 | 88.3 | 8.4 | 3.3 | 94.8 | 2.3 | 2.9 | <0.001 |
| Did your nurses care about you? | 1478 | 99.7 | 96.3 | 1.4 | 2.3 | 96.5 | 1.1 | 2.5 | 96.4 | 1.8 | 1.8 | 0.50 |
| Did your nurses answer your questions? | 1457 | 98.3 | 96.2 | 1.2 | 2.6 | 95.6 | 1.6 | 2.9 | 96.9 | 0.8 | 2.3 | 0.52 |
| Did your nurses spend enough time with you? | 1469 | 99.1 | 93.6 | 1.9 | 4.5 | 93.8 | 1.2 | 5.0 | 94.9 | 1.8 | 3.4 | 0.39 |
| Did your nurses respect your choices? | 1448 | 97.6 | 92.1 | 1.2 | 6.7 | 91.0 | 2.2 | 6.9 | 93.5 | 0.5 | 6.0 | 0.10 |
| Did your nurses help solve your problems? | 1406 | 94.8 | 90.8 | 2.4 | 6.9 | 91.1 | 2.4 | 6.5 | 91.9 | 1.3 | 6.7 | 0.56 |
| Did your nurses talk to you about your problems? | 1359 | 91.6 | 78.7 | 9.8 | 11.6 | 79.1 | 9.9 | 11.0 | 79.8 | 9.8 | 10.4 | 0.95 |
| Did your nurses explain what choices you had? | 1354 | 91.3 | 78.1 | 9.5 | 12.4 | 77.4 | 8.9 | 13.7 | 77.8 | 10.0 | 12.2 | 0.72 |
| Did your nurses treat you with respect? | 1474 | 99.4 | 94.0 | 2.1 | 3.9 | 92.1 | 3.2 | 4.8 | 95.1 | 1.8 | 3.1 | 0.17 |
| Did your nurses support your decisions? | 1392 | 93.9 | 85.9 | 3.4 | 10.8 | 85.6 | 3.6 | 10.8 | 85.8 | 3.8 | 10.4 | 0.97 |
| Were you satisfied with the amount of time the nurse had you wait? | 1401 | 94.5 | 56.0 | 26.0 | 18.0 | 53.4 | 28.7 | 18.0 | 59.8 | 23.4 | 16.8 | 0.13 |
| Did your nurses refer you to the right people if they could not help you? | 1303 | 87.9 | 72.4 | 14.7 | 12.9 | 71.5 | 16.8 | 11.6 | 75.9 | 12.9 | 11.2 | 0.27 |
| Did your nurses speak to you with respect? | 1476 | 99.5 | 95.3 | 1.5 | 3.3 | 95.1 | 1.2 | 3.7 | 94.9 | 1.8 | 3.3 | 0.75 |
| Were you able to open up to your nurses without feeling intimidated? | 1374 | 92.7 | 81.2 | 8.9 | 10.0 | 82.4 | 9.4 | 8.2 | 81.3 | 8.0 | 10.7 | 0.37 |
| Did the receptionist treat you with respect?* | 1253 | 84.5 | 95.5 | 1.8 | 2.8 | 96.7 | 1.9 | 2.5 | 95.6 | 2.1 | 2.4 | 0.97 |
| Did the laboratory staff treat you with respect?* | 1034 | 69.7 | 94.7 | 1.9 | 3.4 | 94.2 | 2.3 | 3.5 | 94.3 | 1.4 | 4.3 | 0.70 |
| Did the pharmacy staff treat you with respect?* | 800 | 53.9 | 95.4 | 0.9 | 3.8 | 94.4 | 0.9 | 4.7 | 96.3 | 0.5 | 3.2 | 0.66 |
| Did the mothers2mothers staff treat you with respect?* | 840 | 56.6 | 97.4 | 0.8 | 1.8 | 98.0 | 0.7 | 1.4 | 97.5 | 0.9 | 1.7 | 0.90 |
| Do you think healthcare workers at this clinic kept your information private? | 793 | 53.5 | 82.7 | 2.5 | 14.8 | 80.7 | 2.1 | 17.2 | 83.1 | 3.7 | 13.2 | 0.30 |
| Do you think healthcare workers here treat people who are HIV positive worse than people who are HIV negative? | 1075 | 72.5 | 32.1 | 39.2 | 28.7 | 32.6 | 37.2 | 30.2 | 28.2 | 43.6 | 28.2 | 0.21 |

* Only participants who had interacted with these clinic staff at their visits were given the opportunity to respond to these questions

**Table 2. Patient feedback survey responses from pregnant women attending antenatal care at seven health facilities in Eswatini by self-reported HIV status, August/September-December 2017 (N = 1,218).**

| Survey question | HIV-negative (N = 710) | | | HIV-positive (N = 508) | | | |
|---|---|---|---|---|---|---|---|
| | Percent Agree | Percent Disagree | Percent Neutral | Percent Agree | Percent Disagree | Percent Neutral | p-value |
| Did your nurses listen to you? | 92.0 | 5.2 | 2.9 | 91.2 | 5.9 | 2.9 | 0.86 |
| Did your nurses care about you? | 97.0 | 1.1 | 1.8 | 96.3 | 1.8 | 2.0 | 0.63 |
| Did your nurses answer your questions? | 95.7 | 1.4 | 2.9 | 97.6 | 1.4 | 1.0 | 0.08 |
| Did your nurses spend enough time with you? | 93.3 | 1.7 | 5.0 | 96.0 | 1.8 | 2.2 | 0.05 |
| Did your nurses respect your choices? | 91.8 | 1.4 | 6.8 | 93.6 | 1.4 | 5.0 | 0.45 |
| Did your nurses help solve your problems? | 90.3 | 1.7 | 8.1 | 91.7 | 3.8 | 4.4 | <0.01 |
| Did your nurses talk to you about your problems? | 77.3 | 9.8 | 12.9 | 83.4 | 9.4 | 7.3 | 0.01 |
| Did your nurses explain what choices you had? | 76.5 | 10.3 | 13.2 | 80.9 | 8.0 | 11.1 | 0.20 |
| Did your nurses treat you with respect? | 93.5 | 1.8 | 4.7 | 94.6 | 3.0 | 2.4 | 0.05 |
| Did your nurses support your decisions? | 83.8 | 2.7 | 13.5 | 90.3 | 3.5 | 6.2 | <0.001 |
| Were you satisfied with the amount of time the nurse had you wait? | 52.6 | 29.9 | 17.5 | 60.5 | 21.9 | 17.6 | <0.01 |
| Did your nurses refer you to the right people if they could not help you? | 70.1 | 16.4 | 13.5 | 74.5 | 14.5 | 11.0 | 0.26 |
| Did your nurses speak to you with respect? | 94.3 | 1.4 | 4.3 | 96.4 | 1.8 | 1.8 | 0.05 |
| Were you able to open up to your nurses without feeling intimidated? | 78.7 | 9.7 | 11.6 | 85.2 | 8.3 | 6.5 | <0.01 |
| Did the receptionist treat you with respect?* | 94.1 | 2.2 | 3.7 | 98.2 | 0.9 | 0.9 | <0.01 |
| Did the laboratory staff treat you with respect?* | 94.9 | 1.9 | 3.2 | 94.6 | 2.4 | 3.0 | 0.870 |
| Did the pharmacy staff treat you with respect?* | 92.9 | 1.2 | 5.9 | 97.8 | 0.6 | 1.6 | 0.006 |
| Did the mothers2mothers staff treat you with respect?* | 97.0 | 1.4 | 1.7 | 97.7 | 0.6 | 1.7 | 0.66 |
| Do you think healthcare workers at this clinic kept your information private? | 80.1 | 2.7 | 17.2 | 84.9 | 2.4 | 12.8 | 0.26 |
| Do you think that healthcare workers here treat people who are HIV positive worse than people who are HIV negative? | 24.5 | 43.5 | 32.1 | 43.9 | 32.9 | 23.2 | <0.0001 |

*Only participants who had interacted with these clinic staff at their visits were given the opportunity to respond to these questions

which increased to 64.6% in the post-period but was not statistically significant. We also measured retention in ANC care at three and six months after the first visit (based on gestational age at ANC entry) and found a significant increase in the proportion of all pregnant women retained at six months, from 60.9% to 72.7% (p = 0.03) and among HIV-negative women from 56.6% to 71.6% (p = 0.02). The increased retention observed in the post-period was not statistically significant among WLHIV.

There are only a small number of studies in which ANC attenders have been asked about their interactions with healthcare providers, most previous studies focus on mistreatment at the time of delivery. One study from rural Ethiopia found that among 288 pregnant women, 89% reported satisfaction with ANC services and 72% reported that providers listened to their problems [24]. Another study from urban Ethiopia found that 96% of ANC attenders reported that they were happy with their visit and 93% felt comfortable with the counseling received for routine ANC HIV testing [25]. A study from Tanzania found that 19.5% of women reported disrespectful care following delivery, and in Nigeria, 98.0% of new mothers reported mistreatment (of any kind) at the 6 week postpartum visit [13,15,26]. While our survey did not ask directly about mistreatment or abuse, our findings suggest that pregnant women at the seven health facilities in Eswatini included in this study felt that the care they received was respectful and that the vast majority of women were satisfied with their interactions with healthcare workers overall.

**Table 3. Characteristics and retention among pregnant women newly entering ANC at two health facilities in Eswatini in the pre (Jan-Feb 2017) and post (Jan-Feb 2018) intervention periods (N = 680).**

| | Pre-period (Jan-Feb 2017) | | | | | | Post-period (Jan-Feb 2018) | | | | | | Pre/post all | Pre/post HIV-negative | Pre/post HIV-positive |
|---|---|---|---|---|---|---|---|---|---|---|---|---|---|---|---|
| | Pre-period all | | HIV-negative | | HIV-positive | | Post-period all | | HIV-negative | | HIV-positive | | p-value | p-value | p-value |
| | N | % | N | % | N | % | N | % | N | % | N | % | | | |
| | 330 | 100.0 | 228 | 33.5 | 102 | 15.0 | 350 | 100.0 | 226 | 33.2 | 124 | 18.2 | | | |
| **HIV status at entry to ANC** | | | | | | | | | | | | | | | |
| Known HIV+ | 60 | 58.8 | | | 60 | 58.8 | 77 | 62.1 | | | 77 | 62.1 | | | 0.2525 |
| Tested HIV+ at first ANC visit | 41 | 40.2 | | | 41 | 40.2 | 42 | 33.9 | | | 42 | 33.9 | | | |
| Tested HIV+ during ANC | 1 | 1.0 | | | 1 | 1.0 | 5 | 4.0 | | | 5 | 4.0 | | | |
| **Age**, median (interquartile range) | 27 (23–31) | | 26 (23–30) | | 30 (26–34) | | 27 (23–32) | | 25 (22–30) | | 30 (25–34) | | 0.41 | 0.86 | 0.47 |
| 14–19 years | 16 | 4.8 | 14 | 6.1 | 2 | 2.0 | 27 | 7.7 | 21 | 9.3 | 6 | 4.8 | 0.75 | 0.53 | 0.83 |
| 20–29 years | 193 | 58.5 | 146 | 64.0 | 47 | 46.1 | 200 | 57.1 | 144 | 63.7 | 56 | 45.2 | | | |
| 30–39 years | 113 | 34.2 | 64 | 28.1 | 49 | 48.0 | 111 | 31.7 | 56 | 24.8 | 55 | 44.4 | | | |
| 40+ years | 8 | 2.4 | 4 | 1.8 | 4 | 3.9 | 12 | 3.4 | 5 | 2.2 | 7 | 5.6 | | | |
| **Gestational age weeks**, median (interquartile range) | 20 (15–25) | | 20 (15–26) | | 20 (16–25) | | 19 (14–24) | | 18 (13–24) | | 19 (15–24) | | 0.01 | 0.03 | 0.23 |
| **Trimester** (8 missing) | | | | | | | | | | | | | | | |
| First Trimester | 67 | 20.3 | 47 | 20.9 | 20 | 20.6 | 95 | 27.1 | 66 | 29.2 | 29 | 23.4 | 0.02 | 0.05 | 0.26 |
| Second Trimester | 191 | 57.9 | 133 | 59.1 | 58 | 59.8 | 201 | 57.4 | 122 | 54.0 | 79 | 63.7 | | | |
| Third Trimester | 64 | 19.4 | 45 | 20.0 | 19 | 19.6 | 54 | 15.4 | 38 | 16.8 | 16 | 12.9 | | | |
| **Attended at least 4 antenatal visits** | 196 | 59.4 | 125 | 54.8 | 71 | 69.6 | 226 | 64.6 | 133 | 58.9 | 93 | 75.0 | 0.19 | 0.44 | 0.35 |
| **Median number ANC visits overall** (range) | 4 (2–5) | | 4 (2–5) | | 5 (3–6) | | 4 (3–5) | | 4 (3–5) | | 5 (4–7) | | 0.10 | 0.13 | 0.76 |
| **Proportion retained at 3 months*** (N = 632) | 223 | 74.6 | 145 | 71.1 | 78 | 82.1 | 256 | 76.9 | 158 | 73.5 | 98 | 83.1 | 0.52 | 0.62 | 0.82 |
| **Proportion retained at 6 months**** (N = 305) | 81 | 60.9 | 52 | 56.5 | 29 | 70.7 | 125 | 72.7 | 83 | 71.6 | 42 | 75.0 | 0.03 | 0.02 | 0.54 |

*among women with first ANC <33 weeks.

**among women with first ANC <19 weeks.

We observed differences in pregnant women's experiences with HFS based on self-reported HIV status, with more WLHIV reporting favorable patient-provider interactions compared to HIV-negative women. However, WLHIV were also more likely to report that HFS treated HIV-positive women worse than HIV-negative women. Our findings identified an interesting discrepancy, namely that almost all WLHIV reported satisfaction with ANC care; however, almost half (44%) of WLHIV also felt that HFS provided worse care than compared to HIV-negative women. While we do not have further data with which to explore this finding, previous work has noted that pregnant women may be particularly vulnerable to HIV-related stigma (both substandard care and perceptions of stigma) [27] and that stigma is a barrier to uptake and retention in care for pregnant WLHIV in PMTCT services [27]. Our study is novel in that few previous studies have compared satisfaction with patient-provider interaction in ANC settings according to HIV status although there are data on satisfaction among WLHIV with PMTCT services. A study from Kenya evaluating an integrated care model found that a higher proportion of HIV-positive women (79%) reported being "very satisfied" with clinical visits compared to 63% of HIV-negative women [28]. In both Ethiopian studies, privacy and confidentiality were the area of greatest dissatisfaction for women, with 26% of women in the

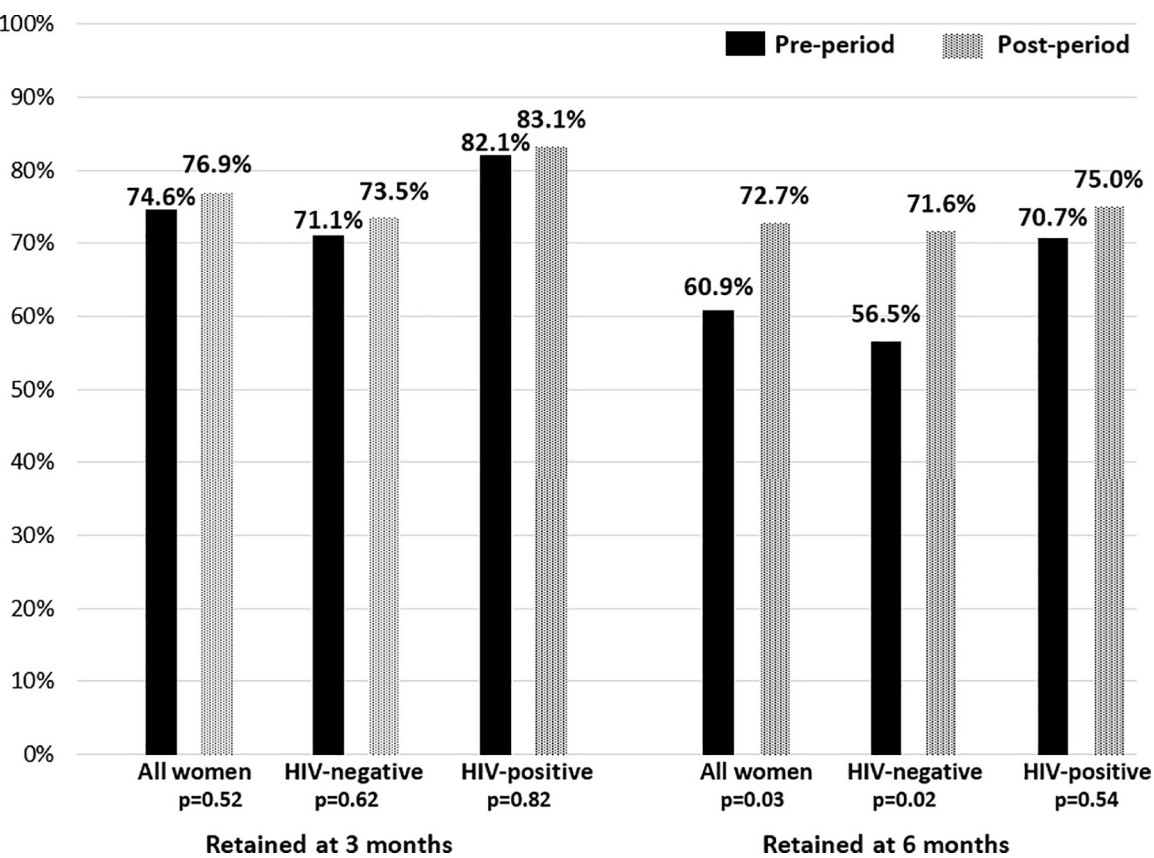

**Fig 2. Women newly enrolled in ANC in the pre- and post-periods.**

urban setting reporting concerns about privacy [24,25]. In comparison, women in Eswatini did not express a similar concern, with only 3% of women reporting that they felt healthcare workers had not kept their information private.

Our findings regarding improved retention and using patient feedback surveys to inform HFS and improve patient-provider relationships in ANC and PMTCT settings showed favorable, but mixed results. While we e observed an increase from the pre- to the post-period in the proportion of HIV-negative pregnant women who were still attending ANC at six months after the first ANC visit increased between the pre- and post-period, we also found overall low uptake of the minimum package of four antenatal care visits (less than 65% among all pregnant women), with no increase observed over time. The disparity in these results may suggest that while total visit attendance may not have increased, after the intervention, more women came back for end of pregnancy care. It is possible that some women came to these larger health facilities for the first visit and then continued care at a different facility; however, we do not have data on transfers and, therefore, cannot account for this. The lack of a clear retention benefit of the intervention is disappointing and suggest that other quality issues such as travel distances, poor facility infrastructure and skill level of providers may have larger impacts on retention [6,7,9]. Unfortunately, our analysis does not provide information about these factors.

The strengths of this analysis include the use of a tablet-based survey to inform rapid quality improvement discussions and the combination of both patient satisfaction data and actual retention outcomes. The patient feedback we report from Eswatini is novel in that it was

collected over multiple months and allowed us to examine change over time; most previous studies have only collected data at one time point [13,24–26]. Our survey also included feedback about patient interactions with not just clinical care providers but also other cadres of essential health workers, including peer supporters, receptionists and laboratory personnel. A final strength is the examination of ANC retention outcomes assessed for both HIV-negative women and WLHIV and which we used to assess the impact of the rapid quality improvement intervention using patient feedback surveys.

There are also some important limitations to both components of the evaluation. As noted above, our survey instrument was focused on patient-provider relationships in the ANC setting and did not ask about structural challenges to ANC attendance which may be important drivers of retention in these services. It is also possible that the high satisfaction levels we measured in the patient surveys was in part driven by response bias as a result of completing the survey within the health facility. Despite assurances of anonymity, women may have felt concerned about their care if they reported dissatisfaction with services. As noted, we found high levels of satisfaction which may have limited our ability to measure the potential impact of this type of intervention, it is possible that in a different setting with lower baseline satisfaction, there might have been a more significant increase in satisfaction. Other limitations include self-reported HIV status from women completing the anonymous surveys which 18% of women did not provide. With regard to the analysis of retention, due to differences in how and where routine data are recorded across sites, we were only able to examine retention outcomes from women attending care at two of the seven project health facilities which used the same national electronic medical record system. The two sites included were larger health facilities than the excluded facilities and it is possible that the outcomes of the retention analysis might have differed with inclusion of data from the smaller sites which limits the generalizability of our findings. Furthermore, our data were limited to what was available in the electronic medical record system which is subject to missingness. It is possible that some women may have transferred to other health facilities for care which we cannot account for.

## Conclusions

Implementation of respectful care policies has been studied previously as a way to improve maternal health services and outcomes [29]. The type of rapid quality improvement intervention we implemented may be useful in improving patient-provider relationships although whether it can improve retention remains unclear.

## Supporting information

**S1 Dataset.**
(XLSX)

## Acknowledgments

### Declarations

The authors would like to thank the pregnant women who completed the patient feedback surveys and all of the HFS who participated in the feedback sessions.

### Ethics approval and consent to participate

The study protocol was reviewed and approved by the Columbia University Medical Center Institutional Review Board (IRB), and the Eswatini National Health Research Review Board.

The study was also reviewed in accordance with and the US Centers for Disease Control and Prevention (CDC) human subjects protections procedures and was determined to be research, but CDC investigators did not interact with human subjects or have access to identifiable data or specimens for research purposes. All women participating in the anonymous patient feedback surveys provided verbal consent. Pregnant women less than 18 years of age were asked to participate and able to provide consent for themselves as pregnant women in Eswatini are considered emancipated minors. Trained data collectors read the consent form to all potential participants which included assurances that participation was voluntary and non-participation would not lead to any differences in the care they received. Verbal, rather than written consent, was taken for those completing the anonymous feedback survey in order to protect women's confidentiality. A waiver of written consent was granted by all of the ethics boards for the use of de-identified electronic medical record data for the retrospective retention analysis.

## Author Contributions

**Conceptualization:** Chloe A. Teasdale, Amanda Geller, Katherine King, Surbhi Modi, Elaine J. Abrams.

**Data curation:** Siphesihle Shongwe, Arnold Mafukidze, Michelle Choy, Bhekinkhosi Magaula, Katharine Yuengling, Katherine King, Eduarda Pimentel De Gusmao.

**Formal analysis:** Chloe A. Teasdale, Katharine Yuengling.

**Funding acquisition:** Amanda Geller, Caroline Ryan, Trong Ao, Tegan Callahan, Surbhi Modi.

**Investigation:** Chloe A. Teasdale, Michelle Choy, Elaine J. Abrams.

**Methodology:** Chloe A. Teasdale, Siphesihle Shongwe, Arnold Mafukidze, Michelle Choy, Bhekinkhosi Magaula, Katherine King, Eduarda Pimentel De Gusmao, Surbhi Modi, Elaine J. Abrams.

**Project administration:** Chloe A. Teasdale, Amanda Geller, Siphesihle Shongwe, Arnold Mafukidze, Michelle Choy, Bhekinkhosi Magaula, Katharine Yuengling, Eduarda Pimentel De Gusmao, Caroline Ryan, Trong Ao, Tegan Callahan, Elaine J. Abrams.

**Supervision:** Chloe A. Teasdale, Elaine J. Abrams.

**Writing – original draft:** Chloe A. Teasdale.

**Writing – review & editing:** Chloe A. Teasdale, Amanda Geller, Siphesihle Shongwe, Arnold Mafukidze, Michelle Choy, Bhekinkhosi Magaula, Katherine King, Eduarda Pimentel De Gusmao, Caroline Ryan, Trong Ao, Tegan Callahan, Surbhi Modi, Elaine J. Abrams.

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
