## [Decision Letter · Decision Letter 0]

4 Feb 2021

PONE-D-20-23800

Patient feedback surveys among pregnant women in Eswatini to improve antenatal care retention

PLOS ONE

Dear Dr. Teasdale,

Thank you for submitting your manuscript to PLOS ONE. After careful consideration, we feel that it has merit but does not fully meet PLOS ONE’s publication criteria as it currently stands. Therefore, we invite you to submit a revised version of the manuscript that addresses the points raised during the review process.

We have received comments from two external reviewers. What we would like you to do now is respond to the reviewers' comments outlined below. In particular, changes we would require for acceptance include: 

Please clearly define the term ANC retention in the Introduction. Outline the theoretical underpinnings to the evaluated intervention in the Introduction and provide a more detailed description of the intervention in the Methods. Rigorously link the conclusions to the intent of the paper, particularly as they relate to the type of rapid quality improvement intervention implemented.With the discrepancy in reviewers' opinions on whether the authors made all data underlying the findings in their manuscript fully available, please ensure data are made accessible in line with PLOS guidelines.

All other suggested changes by the reviewers, we would recommend.

We look forward to receiving your revised manuscript.

Kind regards,

Fiona Lynn, Ph.D.

Academic Editor

PLOS ONE

Reviewers' comments:

Reviewer's Responses to Questions

**Comments to the Author**

1. Is the manuscript technically sound, and do the data support the conclusions?

Reviewer #1: Partly

Reviewer #2: Yes

2. Has the statistical analysis been performed appropriately and rigorously? 

Reviewer #1: Yes

Reviewer #2: Yes

3. Have the authors made all data underlying the findings in their manuscript fully available?

Reviewer #1: Yes

Reviewer #2: No

4. Is the manuscript presented in an intelligible fashion and written in standard English?

Reviewer #1: Yes

Reviewer #2: Yes

5. Review Comments to the Author

Reviewer #1: The authors have submitted a manuscript that will definitely help strengthen the body of knowledge across ANC, PMTCT and the challenges of women completing ANC as a precursor to ensuring that they will deliver with trained health personnel and foster the prevention of maternal/neonatal mortality and morbidity. The conclusions provided however need to be further and rigorously linked to the intent of the paper especially as it relates to the " type of rapid quality improvement intervention"... which was not clearly linked across the paper. The manuscript is well written, clear, concise and in standard English.

I am providing further specific comments below for the authors:

ADDITIONAL/SPECIFIC COMMENTS TO AUTHORS

Line 43: The proportion of women retained at six months increased from 60.9% in the pre-period to 72.7% in the post-period (p=0.03). For HIV-negative women, pre- and post-period six-month retention significantly increased from 56.6% to 71.6% (p=0.02); however, the increase in WLHIV retained at six months from 70.7% (pre-period) to 75.0% (post-period) was not statistically significant (p=0.64).

QUESTION: Are there any qualitative explanations in any form for this? Are there likely underlying stigma issues especially for WLHIV? It is concerning that the authors did not clarify this or even see this itself as a concern.

Lines 47 and 48: The type of rapid quality improvement intervention we implemented may be useful in improving patient-provider relationships although whether it can improve retention remains unclear.

COMMENT: It will be useful for authors to clarify what they think may have then be responsible/suggestive for the significant difference between retention between Women HIV negative and WLHIV given this statement? Furthermore, can also think of what level of work and/or research need to be applied to detail this in future under discussions and conclusion?

METHODS:

Lines 111-115: Quality improvement sessions were held at the end of each month of survey collection with HFS to review patient survey data from their facility. Sessions were led by one nurse from each facility who received training from the study staff to facilitate discussions aimed at identifying strategies to improve patient-provider relationships. During the sessions, HFS reviewed poor scoring questions, identified root causes, solutions, and made goals for improvement over the next month. No additional resources were provided by the study to implement the improvement strategies.

COMMENT: It is unclear nor stated explicitly if there were any specific interventions per provider at these health facilities to either build their knowledge/skills and inter personal communication and counselling to improve attitude and service delivery to the pregnant women. What were ( if any) critical inputs of quality especially re: the health care providers at these facilities that could have led to an expected change?

RESULTS

Lines 153-155: disagreement overall was whether women were satisfied with wait times (56.0% agreed/were satisfied while 26.0% disagreed/unsatisfied). The only significant change in responses from month 1 to 3 was regarding whether nurses listened to women, for which agreement increased from 88.3% to 94.8% (p<0.01).

COMMENTS: What were the likely factors, if any responsible for this? Could this be a perception bias? It could be more useful for the readers if the authors can provide more insight into this.

Lines 161-163: not feel nurses supported their decisions (p<0.001) (Table 2). While a high proportion of all women reported that nurses treated them with respect including 98% of WLHIV, 43.9% of WLHIV also reported feeling that HFS at the facility treated HIV-positive women worse than HIV-negative women (compared to 24.5% of HIV-negative women) (p<0.0001).

COMMENTS: There is a need to clarify this sentence and possibly split into two as there are two measures described here being actual actions versus perception. Maybe helpful to first describe each rather than put both together.

women, pre- and post-period six month retention significantly increased from 56.6% to 71.6% (p=0.02); however, the increase in the proportion of WLHIV retained at six months from 70.7% in the pre-period to 75.0% in the post-period was not statistically significant (p=0.64).

COMMENTS

There is a need to define clearly what retention means in the context of this publication even earlier on in this paper to remove the confusion a round retention as in HIV services. Is retention a common term to describe ANC attendance in RMNCH services?

DISCUSSION

Lines 176-178: The aim of this study was to evaluate an intervention to increase retention of pregnant women in Eswatini through improving patient-provider relationships in ANC and PMTCT settings. The anonymous electronic tablet-based surveys collected from all ANC attenders revealed high levels of satisfaction with their interactions with nurses, peer mentors...

COMMENTS:

There is a need to describe this intervention given that this is now stated to be an evaluation. How long was the intervention for? What were the content of the interventions? Who received the interventions and the expected changes? This is a weak aspect of the paper as the evaluation should have been referenced and fully described both in the introduction and the methods sections earlier on. . It will significantly help to contextualize this paper.

Lines 198-199: Eswatini included in this study felt that the care they received was respectful and that the vast majority of women were satisfied with their interactions with healthcare workers overall.

COMMENTS:

Did the authors considered the different cultural tones to respect across countries? How was this defined in this survey and standardized for comparison with other studies referenced here?

Lines 205-208: While we do not have further data with which to explore this finding, previous work has noted that pregnant women may be particularly vulnerable to HIV-related stigma (both substandard care and perceptions of stigma) and further research is warranted in this area (22).

COMMENTS:

The authors may consider the need to additionally review publications relating to self-stigma by WLHIV and their impact on their perceptions of service providers and services? Stigma is broad with different drivers. It is unclear if the authors have explored self-stigma as an influence on the perceptions of WLHIV in this paper. This is an interesting finding and needs to be fully explored.

Lines 224-226: The lack of a clear retention benefit of the intervention is disappointing and suggest that other quality issues such as travel distances, poor facility infrastructure and skill level of providers may have larger impacts on retention (6, 7, 9). Unfortunately, our analysis does not provide information about these factors.

COMMENTS:

It is important to situate what constituted the retention component of the intervention package being evaluated to enable the reader understand why this is disappointing. Was this intervention being evaluated actively geared towards driving increased ANC attendance and retention? This is why it would be more helpful to describe the intervention in some level of detail.

Reviewer #2: This study evaluates a rapid quality improvement approach using patient feedback from pregnant women attending ANC services with the goal of improving health facility staff service delivery, and by extension, to improve retention. The authors have nicely presented and explained the findings, and they have well-articulated the limitations of the study. They also describe the limited impact of the rapid quality improvement approach and the higher-than-expected retention seen at the Eswatini study sites.

Given the limited findings from this study, however, this manuscript may be better suited for another journal with a focus on maternal health or behavioral health in resource-limited settings. Below are some major and minor technical comments that could strengthen this manuscript:

Major:

1. The bars for Figure 2 appear to be incorrect and it looks like the values for WLHIV and HIV-negative women have been switched around for 3 month and 6 month retention. For example, in the table for WLHIV, 6 month ANC retention rates in the pre-period are 70.7% and post-period are 75%. Figure 2 lists 56.5% and 75% respectively.

2. The baseline retention for WLHIV is relatively high (Figure 2) and is not impacted much by the intervention – it stays relatively similar at the pre or post-period, or at the 3 or 6 month timepoint. What do the authors think is the underlying reason for this high level of retention? For example, could it be due to alignment with ARV pickups or other needed clinical visits? Retention of WLHIV and their infants in PMTCT services is very important, but my concern is that it may be difficult to know the true impact of health provider behavior change (and therefore of this intervention) on retention of WLHIV if their clinic visits are paired with ART pickup.

3. To strengthen the reader’s understanding of the intervention, please add more specific information on the intervention itself in both the abstract and the background section. If there are other examples of recent surveys at ANC that have used this approach, please also reference.

4. The authors should mention in the discussion that the retention results are likely not generalizable to other settings given that they are from two sites.

5. In response to two standard questions about data availability, it does not seem like the questions are fully answered. The authors should review the questions and the PLoS data policy to ensure the data is made accessible in line with PLoS guidelines.

Minor:

1. Line 75 – Is the goal truly to improve relationships with health facility staff? From the manuscript, it seems that the intervention allows staff to use the anonymous patient feedback to modify their behavior/attitudes towards ANC attendees.

2. Line 79 – Is there more recent UNAIDS data for Eswatini that can be referenced?

3. Line 125 – The authors note that the retention data comes from two health facilities – more detail on these is provided later, but it would be useful for the reader to have the information earlier.

4. Line 216 – “Uptake of the minimum package of four antenatal care visits was low and did not appear to change after the intervention” – The authors state earlier that there was some positive change, although it was non-significant (line 168). This statement on line 216 should be clarified.

5. Line 230 - A reference is made to previous studies that collected data at one time point; please include citations here.

6. Line 233 – Add “women” after “HIV-negative”

6. PLOS authors have the option to publish the peer review history of their article (what does this mean?). If published, this will include your full peer review and any attached files.

Reviewer #1: No

Reviewer #2: No

---

## [Author Response · Author response to Decision Letter 0]

16 Feb 2021

Reviewer 1 

Line 43: The proportion of women retained at six months increased from 60.9% in the pre-period to 72.7% in the post-period (p=0.03). For HIV-negative women, pre- and post-period six-month retention significantly increased from 56.6% to 71.6% (p=0.02); however, the increase in WLHIV retained at six months from 70.7% (pre-period) to 75.0% (post-period) was not statistically significant (p=0.64).

QUESTION: Are there any qualitative explanations in any form for this? Are there likely underlying stigma issues especially for WLHIV? It is concerning that the authors did not clarify this or even see this itself as a concern.

Response: Thanks for this input. There is not room in the abstract to discuss these issues which are described more fully in the discussion section. 

Lines 47 and 48: The type of rapid quality improvement intervention we implemented may be useful in improving patient-provider relationships although whether it can improve retention remains unclear.

COMMENT: It will be useful for authors to clarify what they think may have then be responsible/suggestive for the significant difference between retention between Women HIV negative and WLHIV given this statement? Furthermore, can also think of what level of work and/or research need to be applied to detail this in future under discussions and conclusion?

Response: We cannot add this information to the abstract but have added more about this to the introduction and methods section in the paper (see page 5 and 7). 

METHODS:

Lines 111-115: Quality improvement sessions were held at the end of each month of survey collection with HFS to review patient survey data from their facility. Sessions were led by one nurse from each facility who received training from the study staff to facilitate discussions aimed at identifying strategies to improve patient-provider relationships. During the sessions, HFS reviewed poor scoring questions, identified root causes, solutions, and made goals for improvement over the next month. No additional resources were provided by the study to implement the improvement strategies.

COMMENT: It is unclear nor stated explicitly if there were any specific interventions per provider at these health facilities to either build their knowledge/skills and inter personal communication and counselling to improve attitude and service delivery to the pregnant women. What were ( if any) critical inputs of quality especially re: the health care providers at these facilities that could have led to an expected change?

Response: We have added a more complete description of the intervention including the tools used as part of the QI project with HFS. This information has been added to the introduction and methods section (pages 5 and 7).

RESULTS

Lines 153-155: disagreement overall was whether women were satisfied with wait times (56.0% agreed/were satisfied while 26.0% disagreed/unsatisfied). The only significant change in responses from month 1 to 3 was regarding whether nurses listened to women, for which agreement increased from 88.3% to 94.8% (p<0.01).

COMMENTS: What were the likely factors, if any responsible for this? Could this be a perception bias? It could be more useful for the readers if the authors can provide more insight into this.

Response: Unfortunately, we do not have qualitative data to inform the question although we believe that our intervention very likely contributed to this. We do not believe the finding results from a bias in participant’s perceptions. 

Lines 161-163: not feel nurses supported their decisions (p<0.001) (Table 2). While a high proportion of all women reported that nurses treated them with respect including 98% of WLHIV, 43.9% of WLHIV also reported feeling that HFS at the facility treated HIV-positive women worse than HIV-negative women (compared to 24.5% of HIV-negative women) (p<0.0001).

COMMENTS: There is a need to clarify this sentence and possibly split into two as there are two measures described here being actual actions versus perception. Maybe helpful to first describe each rather than put both together.

Response: Thank you for this suggestion. We have revised, “Overall, a high proportion of all women reported that nurses treated them with respect including 98% of WLHIV. A higher proportion of WLHIV (43.9%) compared to HIV-negative women (24.5%) reported feeling that HFS treated HIV-positive women worse than HIV-negative women (p<0.0001).”

women, pre- and post-period six month retention significantly increased from 56.6% to 71.6% (p=0.02); however, the increase in the proportion of WLHIV retained at six months from 70.7% in the pre-period to 75.0% in the post-period was not statistically significant (p=0.64).

COMMENTS

There is a need to define clearly what retention means in the context of this publication even earlier on in this paper to remove the confusion a round retention as in HIV services. Is retention a common term to describe ANC attendance in RMNCH services?

Response: We present data on both ANC attendance and adherence at 6 month; both are defined in the methods section, (line 161-163) “We estimated the median number of ANC visits among all pregnant women and the proportion who attended at least four ANC visits.” (line 164-166) “We also calculated the proportion of all pregnant women retained in ANC care at three and six months after their first ANC visit among those with a first ANC visit <33 weeks gestation and <19 weeks, respectively (a one-month window before and after was applied for each retention endpoint).”

DISCUSSION

Lines 176-178: The aim of this study was to evaluate an intervention to increase retention of pregnant women in Eswatini through improving patient-provider relationships in ANC and PMTCT settings. The anonymous electronic tablet-based surveys collected from all ANC attenders revealed high levels of satisfaction with their interactions with nurses, peer mentors...

COMMENTS:

There is a need to describe this intervention given that this is now stated to be an evaluation. How long was the intervention for? What were the content of the interventions? Who received the interventions and the expected changes? This is a weak aspect of the paper as the evaluation should have been referenced and fully described both in the introduction and the methods sections earlier on. It will significantly help to contextualize this paper.

Response: We have added a more complete description of the intervention including the tools used as part of the QI project with HFS. This information has been added to the introduction and methods section (pages 5 and 7).

Lines 198-199: Eswatini included in this study felt that the care they received was respectful and that the vast majority of women were satisfied with their interactions with healthcare workers overall.

COMMENTS:

Did the authors considered the different cultural tones to respect across countries? How was this defined in this survey and standardized for comparison with other studies referenced here?

Response: Thank you for this question, it is important. As described in the methods section (page 6), we have the following information, “The patient feedback survey was designed for both HIV-positive and HIV-negative pregnant women based on the Patient Provider Relationship Scale (PPRS), a validated instrument developed in South Africa for the ANC and PMTCT context that specifically focuses on soliciting patient feedback regarding providers listening to, caring about, and respecting them (11). The PPRS was translated into siSwati and additional questions were added (based on existing questions) to ask about interactions with non-clinical providers (mentor mothers, receptionists, and lab technicians) among the subset of women reporting interactions with these staff at the visit.” 

Lines 205-208: While we do not have further data with which to explore this finding, previous work has noted that pregnant women may be particularly vulnerable to HIV-related stigma (both substandard care and perceptions of stigma) and further research is warranted in this area (22).

COMMENTS:

The authors may consider the need to additionally review publications relating to self-stigma by WLHIV and their impact on their perceptions of service providers and services? Stigma is broad with different drivers. It is unclear if the authors have explored self-stigma as an influence on the perceptions of WLHIV in this paper. This is an interesting finding and needs to be fully explored.

Response: Thank you for this suggestion. Although this is not the focus of our study and we do not have data to directly comment on self-stigma, we have added to our discussion including adding a new reference (page 12), “While we do not have further data with which to explore this finding, previous work has noted that pregnant women may be particularly vulnerable to HIV-related stigma (both substandard care and perceptions of stigma) (27) and that stigma is a barrier to uptake and retention in care for pregnant WLHIV in PMTCT services.(27)”

Lines 224-226: The lack of a clear retention benefit of the intervention is disappointing and suggest that other quality issues such as travel distances, poor facility infrastructure and skill level of providers may have larger impacts on retention (6, 7, 9). Unfortunately, our analysis does not provide information about these factors.

COMMENTS:

It is important to situate what constituted the retention component of the intervention package being evaluated to enable the reader understand why this is disappointing. Was this intervention being evaluated actively geared towards driving increased ANC attendance and retention? This is why it would be more helpful to describe the intervention in some level of detail.

Response: Thank you for drawing out attention to this. As noted, we have added a more complete description of the intervention including the aims of the project to the introduction and methods section (pages 5 and 7).

Review 2

1. The bars for Figure 2 appear to be incorrect and it looks like the values for WLHIV and HIV-negative women have been switched around for 3 month and 6 month retention. For example, in the table for WLHIV, 6 month ANC retention rates in the pre-period are 70.7% and post-period are 75%. Figure 2 lists 56.5% and 75% respectively.

Response: Thank you very much for noting this issue – we have fixed Figure 2 and it is now correct. 

2. The baseline retention for WLHIV is relatively high (Figure 2) and is not impacted much by the intervention – it stays relatively similar at the pre or post-period, or at the 3 or 6 month timepoint. What do the authors think is the underlying reason for this high level of retention? For example, could it be due to alignment with ARV pickups or other needed clinical visits? Retention of WLHIV and their infants in PMTCT services is very important, but my concern is that it may be difficult to know the true impact of health provider behavior change (and therefore of this intervention) on retention of WLHIV if their clinic visits are paired with ART pickup. 

Response: This is an interesting point. We agree that the baseline retention is high which is consistent with previous studies from Eswatini showing low vertical transmission (as mentioned in introduction and discussion). We did not collect data for HIV+ women related to ART pick-up however per Eswatini national guidelines, ART services are integrated into ANC care so it is likely that most HIV+ women would have been able to pick up their ART at ANC visits. 

3. To strengthen the reader’s understanding of the intervention, please add more specific information on the intervention itself in both the abstract and the background section. If there are other examples of recent surveys at ANC that have used this approach, please also reference.

Response: Thank you for this input. We have added more information about the intervention to the introduction and methods, including a clarification of the dates and which data were included (see pages 5, 6 and 7-8). 

4. The authors should mention in the discussion that the retention results are likely not generalizable to other settings given that they are from two sites.

Response: We have added this to the limitations (line 255) 

5. In response to two standard questions about data availability, it does not seem like the questions are fully answered. The authors should review the questions and the PLoS data policy to ensure the data is made accessible in line with PLoS guidelines.

Response: We have revised our submission to include the anonymous patient feedback survey data which will be made publicly available. Unfortunately, we are not permitted to make the patient level data that were used for the retention analysis publicly available and these must remain available by request. According to the approval study protocol, the routinely-collected electronic medical record data used for that part of the study belong solely to the Eswatini Ministry of Health who have not given permission for them to be made publicly available. 

Minor:

1. Line 75 – Is the goal truly to improve relationships with health facility staff? From the manuscript, it seems that the intervention allows staff to use the anonymous patient feedback to modify their behavior/attitudes towards ANC attendees.

Response: Yes, the goal was to provide feedback to the HF staff so that they could use that feedback to improve their relationships with the ANC attendees. Our intervention was limited in time and scope, it was considered a rapid quality improvement initiative and was designed to be an intervention that could be feasibly mounted with limited resources. We have added more information about the intervention to the methods section (see page 5 and 7). 

2. Line 79 – Is there more recent UNAIDS data for Eswatini that can be referenced?

Response: There are newer data however the 2018 data are closer to when the project was conducted. 

3. Line 125 – The authors note that the retention data comes from two health facilities – more detail on these is provided later, but it would be useful for the reader to have the information earlier.

Response: We have added the following to the methods section (page 8), “The two health facilities with available electronic medical records included a large government hospital and a clinic run by a non-governmental organization both located in the capital city of Mbabane.”

4. Line 216 – “Uptake of the minimum package of four antenatal care visits was low and did not appear to change after the intervention” – The authors state earlier that there was some positive change, although it was non-significant (line 168). This statement on line 216 should be clarified.

Response: Thank you for noting this, we have revised this section to more clearly describe the mixed results in our study (see page 10): “Our findings regarding improved retention and using patient feedback surveys to inform HFS and improve patient-provider relationships in ANC and PMTCT settings showed favorable, but mixed results. While we observed an increase from the pre- to the post-period in the proportion of HIV-negative pregnant women who were still attending ANC at six months after the first ANC visit increased between the pre- and post-period, we also found overall low uptake of the minimum package of four antenatal care visits (less than 65% among all pregnant women), with no increase observed over time. The disparity in these results may suggest that while total visit attendance may not have increased, after the intervention, more women came back for end of pregnancy care.”

5. Line 230 - A reference is made to previous studies that collected data at one time point; please include citations here.

Response: Citations have been added 

6. Line 233 – Add “women” after “HIV-negative”

Response: Thanks, added

---

## [Editor Report · Decision Letter 1]

4 Mar 2021

Patient feedback surveys among pregnant women in Eswatini to improve antenatal care retention

PONE-D-20-23800R1

Dear Dr. Teasdale,

We’re pleased to inform you that your manuscript has been judged scientifically suitable for publication and will be formally accepted for publication once it meets all outstanding technical requirements.

Kind regards,

Fiona Lynn, Ph.D.

Academic Editor

PLOS ONE

---

## [Editor Report · Acceptance letter]

9 Mar 2021

PONE-D-20-23800R1 

Patient feedback surveys among pregnant women in Eswatini to improve antenatal care retention 

Dear Dr. Teasdale:

I'm pleased to inform you that your manuscript has been deemed suitable for publication in PLOS ONE. Congratulations! Your manuscript is now with our production department. 

Kind regards, 

on behalf of

Dr. Fiona Lynn 

Academic Editor

PLOS ONE